# GS4: GENERALIZABLE SPARSE SPLATTING SEMANTIC SLAM

## ABSTRACT

Traditional SLAM algorithms excel at camera tracking, but typically produce incomplete and low-resolution maps that are not tightly integrated with semantics prediction. Recent work integrates Gaussian Splatting (GS) into SLAM to enable dense, photorealistic 3D mapping, yet existing GS-based SLAM methods require per-scene optimization that is slow and consumes an excessive number of Gaussians. We present GS4, the first *generalizable* GS-based semantic SLAM system. Compared with prior approaches, GS4 runs 10× faster, uses 10× fewer Gaussians, and achieves state-of-the-art performance across color, depth, semantic mapping and camera tracking. From an RGB-D video stream, GS4 incrementally builds and updates a set of 3D Gaussians using a feed-forward network. First, the Gaussian Prediction Model estimates a sparse set of Gaussian parameters from input frame, which integrates both color and semantic prediction with the same backbone. Then, the Gaussian Refinement Network merges new Gaussians with the existing set while avoiding redundancy. Finally, we propose to optimize GS for only 1-5 iterations that corrects drift and floaters when significant pose changes are detected. Experiments on the real-world ScanNet benchmark demonstrate state-of-the-art semantic SLAM performance, with strong generalization capability shown through zero-shot transfer to the NYUv2 and TUM RGB-D datasets.

## 1 INTRODUCTION

Simultaneous Localization and Mapping (SLAM) is a long-standing challenge in computer vision, aiming to reconstruct a 3D map of an environment while simultaneously estimating camera poses from a video stream. Semantic visual SLAM extends this goal by producing dense maps enriched with semantic labels, enabling applications in autonomous driving, AR/VR, and robotics. By combining geometric reconstruction with object-level understanding, semantic SLAM provides rich 3D spatial and semantic information that allows robots and other systems to navigate and interact with their surroundings more effectively.

Traditional visual SLAM systems consist of several independent components, including keypoint detection, feature matching, and bundle adjustment (Mur-Artal et al., 2015; Mur-Artal & Tardós, 2017; Campos et al., 2021) Their scene representations are typically low-resolution voxels, which limit geometric detail. Thus, although these systems generally provide accurate camera localization, they struggle to generate dense, high-quality 3D maps, which are required for robotics applications such as mobile manipulation. Recent advances in differentiable rendering (Mildenhall et al., 2020; Kerbl et al., 2023a) introduces new options for scene representation in visual SLAM. For example, neural scene representations such as Neural Radiance Fields (NeRF) (Mildenhall et al., 2020) have been successfully adopted in SLAM frameworks (Sucar et al., 2021; Zhu et al., 2022b; Johari et al., 2023); however, NeRF requires hours of per-scene optimization, making it computationally expensive and forcing a trade-off between reconstruction quality and training cost.

Recently, 3D Gaussians have emerged as a powerful 3D scene representation, offering fast, differentiable, and high-quality rendering capabilities (Kerbl et al., 2023b). Leveraging these advantages, Gaussian-based representations have proven highly effective for SLAM systems (Keetha et al., 2024; Matsuki et al., 2024). However, existing approaches still rely on test-time, gradient-based optimization to estimate 3D Gaussians for each scene independently, which is computationally expensive and unsuitable for real-time applications. In addition, these methods depend on heuristic Gaussian den-

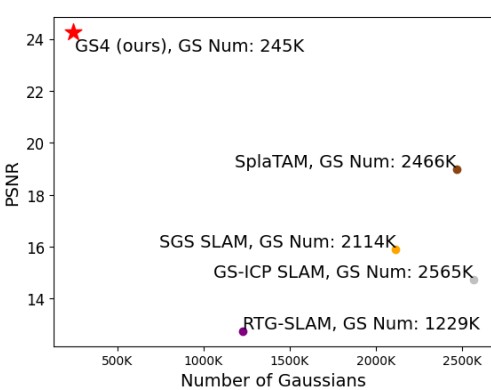

Figure 1: Comparison of PSNR with respect to number of Gaussians across Gaussian Splatting SLAM algorithms (over an average of $2,680$ frames in the 6 testing scenes of ScanNet). Our method achieves state-of-the-art performance with much fewer Gaussians. GS Num represents the number of 3D Gaussians in the scene after mapping is complete.

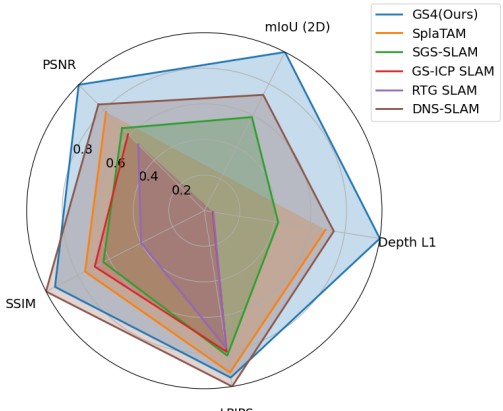

Figure 2: A radar chart comparing rendering metrics. We normalize each metric independently, values closer to the outer edge indicate better performance.

sification and pruning strategies (Kerbl et al., 2023b), often producing overly dense representations that fail to scale to large, real-world environments.

In this paper, we propose **GS4** (**G**eneralizable **S**parse **S**platting **S**emantic **S**LAM), the first generalizable Gaussian-splatting–based SLAM system, which directly predicts 3D semantic Gaussians using a learned feed-forward network, eliminating the need for expensive per-scene optimization. By integrating an image recognition backbone, GS4 jointly reconstructs geometry, color, and semantic labels of the environment without relying on any external semantic-segmentation modules.

GS4 begins with the Gaussian Prediction Model that infers a sparse set of 3D semantic Gaussians from each incoming RGB-D frame in a feed-forward manner. Next, the Gaussian Refinement Network integrates these newly predicted Gaussians with the evolving 3D map, replacing the hand-crafted heuristics traditionally used for Gaussian densification and pruning. This learned refinement strategy yields a compact representation with an order-of-magnitude fewer Gaussians than competing methods. Finally, after the global localization (bundle adjustment) step from the camera tracking module updates camera poses and Gaussian locations, we perform a lightweight few-iteration (only $1{\sim}5$) optimization of Gaussian parameters to preserve rendering fidelity and mitigate the "floater" artifacts common in feed-forward GS approaches.

We demonstrate that GS4 achieves state-of-the-art performance across all key metrics in localization, mapping, and segmentation on the real-world benchmark ScanNet (Fig. 2), while using only $\sim 10\%$ of Gaussians compared to prior GS SLAM methods (Fig. 1). Furthermore, we highlight the generalization capability of our system via zero-shot transfer to the NYUv2 and TUM RGB-D datasets, which, to the best of our knowledge, is the first demonstration of **zero-shot semantic SLAM** generalization in a modern neural SLAM system.

In summary, our contributions are as follows:

- We propose GS4, the first generalizable Gaussian splatting semantic SLAM approach on monocular RGB-D sequences. Results showed that GS4 obtains state-of-the-art on real ScanNet scenes and also zero-shot generalizes to the real NYUv2 and TUM RGB-D datasets without any fine-tuning.

- Our proposed Gaussian refinement network effectively merges Gaussians from different frames into a 3D representation, while significantly reducing the number of Gaussians required to represent a scene to only $10\% - 25\%$ of prior work.

- Our proposed few-iteration Gaussian optimization significantly improves reconstruction quality with a small additional computational cost.

## 2 RELATED WORK

**Traditional SLAM:** Early visual SLAM methods (Mur-Artal et al., 2015) demonstrated robust localization through effective keypoint detection and matching, which resulted in sparse 3D reconstructions. While these approaches provided reliable localization, the sparse nature of the reconstructed maps limited their utility in applications requiring detailed 3D maps. To address this issue, dense visual SLAM (Kerl et al., 2013; Czarnowski et al., 2020) focused on constructing detailed maps to support applications like augmented reality (AR) and robotics. Prior methods (Canelhas et al., 2013; Dai et al., 2017b; Newcombe et al., 2011; Bylow et al., 2013; Whelan et al., 2013; Prisacariu et al., 2017) employ representations based on Signed Distance Fields (SDF), rather than relying on sparse representations such as point clouds or grids. However, these approaches often suffer from over-smoothed reconstruction, failing to capture fine details crucial for certain tasks.

**NeRF-based SLAM:** Neural Radiance Fields (NeRF) (Mildenhall et al., 2020) gained popularity as a 3D scene representation due to its ability to generate accurate and dense reconstructions. NeRF employs Multi-Layer Perceptron (MLP) to encode scene information and performs volume rendering by querying opacity and color along pixel rays. Methods such as iMAP (Sucar et al., 2021), NICE-SLAM (Zhu et al., 2022b), and ESLAM (Johari et al., 2023) incorporate this implicit scene representation into SLAM, leveraging NeRF's high-fidelity reconstructions to improve both localization and mapping. DNS-SLAM (Li et al., 2023) further incorporates semantic information into the framework. However, the volumentric rendering process in NeRF is costly, often requiring trade-offs such as limiting the number of pixels during rendering, These trade-offs, while improving efficiency, may compromise the system's accuracy in both localization and mapping.

**GS-based SLAM:** 3D Gaussian Splatting (3DGS) (Kerbl et al., 2023b) employs splatting rasterization instead of ray marching. This approach iterates over 3D Gaussian primitives rather than marching along rays, resulting in a more expressive and efficient representation capable of capturing high-fidelity 3D scenes with significantly faster rendering speed. Hence, GS-based SLAM systems achieve improved accuracy and speed in dense scene reconstruction. SplaTAM (Keetha et al., 2023) introduces silhouette-guided rendering to support structured map expansion, enabling efficient dense visual SLAM. Gaussian Splatting SLAM (Matsuki et al., 2024) integrates novel Gaussian insertion and pruning strategies, while GS-ICP SLAM (Ha et al., 2024) and RTG-SLAM (Peng et al., 2024) combine ICP with 3DGS to achieve both higher speed and superior map quality. Expanding upon these advancements, SGS-SLAM (Li et al., 2024), OVO-SLAM (Martins et al., 2024), and SemGauss-SLAM (Zhu et al., 2025) extend 3D Gaussian representations to include semantic scene understanding. However, existing GS-based SLAM methods employ per-scene optimization, requiring iterative refinement of Gaussians initialized from keyframes through rendering supervision. As a result, they all rely on additional segmentation models to predict semantic labels for each image, creating computational overhead.

**Feed-forward Models for GS:** Recent research has introduced feed-forward approaches for scene-level 3DGS reconstruction using generalizable models (Charatan et al., 2024; Chen et al., 2024; Liu et al., 2025). Unlike previous methods, GS-LRM (Zhang et al., 2024) avoids specialized 3D structural designs, instead using a transformer to achieve state-of-the-art results. However, to the best of our knowledge, feed-forward models have been applied on a small number of images and have not yet been introduced in GS-based semantic SLAM approaches with thousands of frames.

## 3 METHODS

In this section, we describe our proposed SLAM approach. We first provide a brief overview of Gaussian Splatting, then detail our Gaussian prediction network and Gaussian refinement network. Finally, we explain how these networks are utilized within the entire SLAM system.

### 3.1 GAUSSIAN SPLATTING

We represent a 3D map using a set of anisotropic 3D Gaussians. Each Gaussian $G_i$ is characterized by RGB color $c_i \in \mathbb{R}^3$, center position $\mu_i \in \mathbb{R}^3$, scale $s_i \in \mathbb{R}^3$, quaternion $r_i \in \mathbb{R}^4$, opacity $o_i \in \mathbb{R}$ and semantic class vector $v_i^{\text{class}} \in \mathbb{R}^N$, where N is the number of classes.

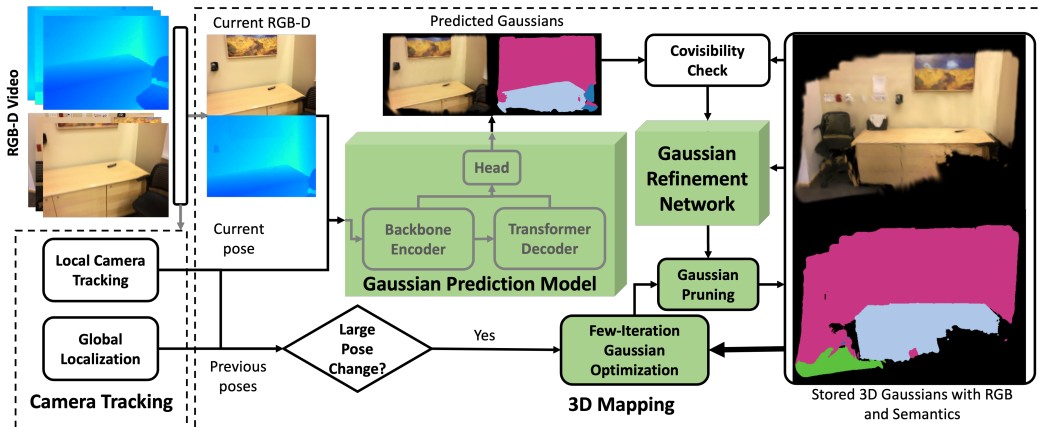

Figure 3: **Overview of the SLAM System.** At each timestep, the system receives an RGB-D frame as input. The tracking system performs local camera tracking and global localization to determine the current frame's pose and correct previous pose errors. Our 3D mapping process comprises three main components: **1) Gaussian Prediction (Sec 3.2.1):** Utilizing the current frame's RGB-D data, the Gaussian Prediction Model estimates the parameters and semantic labels for all Gaussians in the current frame; **2) Gaussian Refinement (Sec 3.2.2):** Both newly added Gaussians and those in the existing semantic 3D map are refined using the Gaussian Refinement Network to ensure that the combined set of Gaussians accurately represents the scene. A covisibility check ensures that only non-overlapping Gaussians are integrated into the existing 3D map. Post-refinement, the transparent Gaussians are pruned; **3) Few-Iteration Gaussian Optimization (Sec 3.3.2):** If significant pose corrections happen, few-iteration Gaussian optimization is performed to update the 3D map's Gaussians, ensuring consistency with the revised camera poses. (Best viewed in color)

The rendering process is defined as:

$$Q_p = \sum_{i \in N} q_i \alpha_i \prod_{j=1}^{i-1} (1 - \alpha_j),$$

where $Q_p$ is a quantity of a pixel $p$ to be rendered, which can be color, depth or semantic label, and $q_i$ is that quantity of the $i$-th 3D Gaussian, while $\alpha_i$ is its visibility, computed from opacity and covariance parameters (determined by rotation and scale). Following Keetha et al. (2024), We also render a silhouette image to determine visibility $S_p = \sum_{i \in N} \alpha_i \prod_{j=1}^{i-1} (1 - \alpha_j)$.

## 3.2 GAUSSIAN PREDICTION AND REFINEMENT

Our proposed Gaussian prediction network (Fig. 3) takes RGB-D images as input and predicts 3D Gaussian parameters. Importantly, the backbone generates features that can predict semantic labels (e.g. trained from 2D segmentation tasks), enabling the rendering of photometric, geometric, and semantic views. Next, the Gaussian refinement network processes Gaussians predicted from a new frame and learns to merge them with the 3D scene representation computed from prior frames.

### 3.2.1 BACKBONE FOR GAUSSIAN PREDICTION

We train a transformer model to regress 3D GS parameters from an image with a known camera pose (from tracking, described in Sec. 3.3.1), while simultaneously assigning semantic labels to these 3D Gaussians. We start with a pre-trained 2D image segmentation model such as Mask2Former (Cheng et al., 2022) or AutoFocusFormer (Ziwen et al., 2023), which encodes an image into encoder tokens $f_{enc}^l$ and decoder tokens $f_{dec}^l$ (from their image decoder) at several progressively downsampled levels $l = 1, \ldots, L$, with $L = 4$ usually. We concatenate an RGB image $I \in R^{H \times W \times 3}$ and a depth image $D \in R^{H \times W \times 1}$ resulting in a 4-channel feature map that is fed into the model:

$$\{f_{\text{enc}_i}^l, f_{\text{dec}_i}^l\}_{i=1:N_{\text{token}}} = \text{Backbone}([I, D])$$

where $N_{\text{token}}$ denotes the total number of prediction tokens per image. The variable $l$ represents the network level at which Gaussians are predicted. If the second level is chosen, the feature map usually has a spatial resolution of $H/8 \times W/8$, resulting in $N_{\text{token}} = HW/64$ tokens per image.

**Processing Prediction Tokens with Transformer.** Given the selected prediction level, we concatenate the encoder features $f_{\text{enc}_i}$ and decoder features $f_{\text{dec}_i}$ for each token $i$, and process the resulting tokens using local-attention transformer layers in the image space to obtain the final $f_i$ features for the $i$-th token, integrating information from both the encoder and decoder.

**Decoding Prediction Tokens to Gaussians.** Each output token's features, $f_i$, from the transformer layers are decoded into Gaussian parameters using Multi-Layer Perceptron (MLP):

$$\{\Delta x_i, \Delta y_i, \Delta d_i, \Delta c_i, s_i, r_i, o_i\} = \text{MLP}(f_i),$$

Here, $\Delta x_i$ and $\Delta y_i$ represent the offsets from the 2D position $(x_i, y_i)$ of the token $f_i$ in the image space, while $\Delta d_i$ is the offset for the noisy depth $d_i$ obtained from the depth image. These offsets are added to the original values, which are then backprojected into 3D space using the intrinsic and extrinsic parameters of the camera, yielding the 3D center position $\mu_i$. Similarly, $\Delta c_i$ represents the offset for the RGB values, obtained from the downsampled image, where each token corresponds to a single pixel. Adding the offset to this value yields the final RGB color for each Gaussian. Besides Gaussian parameters, the mask decoder head in the segmentation model predicts token-level semantic segmentation label vector $v_i^{\text{class}}$ for the input image, which we then assign to the associated Gaussian. Finally, each Gaussian is assigned the final feature vector $f_i$ of its corresponding token for the subsequent Gaussian refinement stage.

To supervise the prediction of semantic segmentation, we follow the setup in Mask2Former (Cheng et al., 2022). We denote the corresponding segmentation loss by $\mathcal{L}_{seg}$. In addition, we render images at $M$ supervision views—comprising the current input view and randomly selected novel views that overlap with the current input—using the predicted Gaussians from the current input, and minimize RGB-D and semantic rendering loss. For novel view supervision, we focus solely on areas visible in the input view, ensuring that the optimization process focuses on regions consistently observed across both input and novel views. We explain the loss functions used during training below.

**RGB Rendering Loss.** Following previous work Zhang et al. (2024); Ziwen et al. (2024), we use a combination of the Mean Squared Error (MSE) loss and Perceptual loss: $\mathcal{L}_{rgb} = \frac{1}{M} \sum_{v=1}^{M} \left( \text{MSE}\left(I_v^{gt}, I_v^{pre}\right) + \lambda \cdot \text{PER}\left(I_v^{gt}, I_v^{pre}\right) \right)$, where $\lambda$ is the weight for the perceptual loss.

**Depth Rendering Loss.** For depth images, we use L1 loss: $\mathcal{L}_d = \frac{1}{M} \sum_{v=1}^{M} \text{L1}\left(D_v^{gt}, D_v^{pre}\right)$.

**Semantic Rendering Loss.** For semantic rendering, we use the cross entropy loss: $\mathcal{L}_{Sem} = \frac{1}{M} \sum_{v=1}^{M} \text{Cross\_Entropy}\left(\text{Sem}_v^{gt}, \text{Sem}_v^{pre}\right)$. where the rendered semantic image has N channels, each corresponding to a different semantic category.

**Overall Training Loss.** Our total loss comprises multiple rendering losses and the segmentation loss $\mathcal{L}_{seg}$: $\mathcal{L} = \lambda_{rgb} \cdot \mathcal{L}_{rgb} + \lambda_d \cdot \mathcal{L}_d + \lambda_{Sem} \cdot \mathcal{L}_{Sem} + \mathcal{L}_{seg}$, where we use $\lambda_{rgb} = 1.0$, $\lambda_d = 1.0$ and $\lambda_{Sem} = 0.1$.

### 3.2.2 GAUSSIAN REFINEMENT NETWORK

The previous subsection predicts Gaussian parameters from a single frame. In our SLAM system, as new frames arrive, we insert Gaussians from the frame into unmapped regions of the current 3D reconstruction. We perform co-visibility, which involves rendering a silhouette image for the new frame to identify the regions where new Gaussians should be inserted. To ensure that the combined set of Gaussians accurately represents the scene, we propose a novel Gaussian Refinement Network to refine both the existing Gaussians in the 3D map and the newly added ones, enabling their effective merging. The input to the network includes the features $f_i$ and 3D positions $\mu_i \in \mathbb{R}^3$ of all Gaussians from the 3D map that are visible in the new frame, as well as Gaussians from the new frame. We process these using several local-attention transformer layers with 3D neighborhoods in the world coordinate system to fuse and update the features for each Gaussian. Subsequently, MLP layers predict updates $\Delta c_i \in \mathbb{R}^3$, $\Delta s_i \in \mathbb{R}^3$, $\Delta r_i \in \mathbb{R}^4$ and $\Delta o_i \in \mathbb{R}$ for each Gaussian. These updates refine the Gaussians to accurately render both current and previous views. To supervise the network, we render the current view along with previous overlapping views. The total training loss is:

$$\mathcal{L}_{merge} = \lambda_{rgb} \cdot \mathcal{L}_{rgb} + \lambda_d \cdot \mathcal{L}_d + \lambda_{Sem} \cdot \mathcal{L}_{Sem} \tag{1}$$

where we use $\lambda_{rgb} = 1.0$, $\lambda_d = 1.0$ and $\lambda_{Sem} = 0.1$. After Gaussian refinement, we prune Gaussians whose updated opacity falls below 0.005, effectively removing those that have become

unimportant after merging. These merging-pruning steps lead to a significantly reduced number of Gaussians in the final 3D map with little performance impact.

During testing time, we introduce a threshold $U$ to manage the uncertainty of each Gaussian. Once a Gaussian has been updated $U$ times by the refinement network, we consider its uncertainty sufficiently reduced and exclude them from further updates. We set $U = 8$ in our experiments.

## 3.3 THE SLAM SYSTEM

An overview of the system is summarized in Fig. 3. The system always maintains a set of 3D Gaussians representing the entire scene. For each new RGB-D image, the Gaussian prediction network predicts 3D Gaussian parameters, which can be rendered into high-fidelity color, depth, and semantic images. The Gaussian refinement network refines both the existing Gaussians in the 3D map and the newly added ones to accurately render both current and previous views. During testing, we occasionally run few-iteration test-time optimization and refine 3D Gaussians in the map to reflect camera pose updates from loop closure and bundle adjustment in the tracking module.

### 3.3.1 TRACKING AND GLOBAL BUNDLE ADJUSTMENT

For camera tacking in our SLAM system, we adopt a tracker used in GO-SLAM (Zhang et al., 2023) which is an enhanced version of DROID-SLAM's tracking module (Teed & Deng, 2021). It first predicts motion in every frame. In local camera tracking, a keyframe is initialized when sufficient motion is detected, and loop closure (LC) is performed. Meanwhile, global localization performs full bundle adjustment (BA) for real-time global refinement once the system contains more than 25 keyframes. Both LC and BA help address the problem of accumulated errors and drift that can occur during the localization process.

### 3.3.2 FEW-ITERATION GAUSSIAN OPTIMIZATION

Loop closure and bundle adjustment are essential components in SLAM systems, employed to correct accumulated drift and adjust the camera poses of previous frames. However, these adjustments can cause Gaussians inserted based on earlier, uncorrected poses to misalign with the scene, leading to inaccurate rendering and mapping. It is crucial to implement a mechanism that updates the Gaussians in the 3D map following pose corrections. To address this issue, we propose using rendering-based optimization to update the Gaussian parameters $\mu_i \in \mathbb{R}^3$, $\mathbf{S} \in \mathbb{R}^3$, $\mathbf{Q} \in \mathbb{R}^4$ and $o_i \in \mathbb{R}$ with only a few iterations. We render RGB-D images for the top-k frames, selected based on significant pose changes. This approach maintains the consistency of the 3D map with updated camera poses. To enhance the efficiency of this optimization, we employ the batch rendering technique from Ye et al. (2024). We omit semantic image rendering to improve system efficiency. For few-iteration optimization, we add a SSIM term to the RGB loss, following Kerbl et al. (2023b):

$$\mathcal{L}_{opt} = \frac{1}{M} \sum_{i'=1}^{M} \left( \lambda_{rgb} \cdot \left( (1-\lambda) \cdot \text{L1} \left( I_{i'}^{gt}, I_{i'}^{pre} \right) + \lambda \left( 1 - \text{SSIM}(I_{i'}^{gt}, I_{i'}^{pre}) \right) \right) + \lambda_d \cdot \text{L1}(D_{i'}^{gt}, D_{i'}^{pre}) \right) \quad (2)$$

where $\lambda$ is set to 0.2 for all experiments.

## 4 EXPERIMENTS

### 4.1 EXPERIMENTAL SETUP

**Training Settings.** We train our Gaussian prediction and refinement networks **entirely** on RGB-D videos from the real ScanNet datasets We exclude the six standard SLAM test scenarios and use all remaining training and validation scenes, supervising with 20 common semantic classes. We adopt AutoFocusFormer (Ziwen et al., 2023) as the backbone for both Mask2Former and Gaussian prediction, using the second stage of the backbone as the prediction stage. Additionally, we experiment with Swin Transformer (Liu et al., 2021) as an alternative backbone. The detailed results are provided in the appendix. Following the low-to-high resolution curriculum of (Ziwen et al., 2024), we train the Gaussian prediction network in three stages with input resolutions of 256×256, 480×480, and 640×480. In the first two stages, images are resized such that the shorter side is 256 or 480 pixels and then center-cropped to a square. For the refinement network, which processes multiple

consecutive frames, we adopt a progressive training schedule: beginning with two frames, then four, and finally eight.

**Datasets.** During testing, we evaluate our method on six real-world scenes on ScanNet (Dai et al., 2017a). Additionally, we perform zero-shot experiments on real scenes from NYUv2 (Nathan Silberman & Fergus, 2012) and TUM RGB-D (Sturm et al., 2012a).

**Metrics.** We use PSNR, Depth-L1 (Zhu et al., 2022a), SSIM (Wang et al., 2004), and LPIPS (Zhang et al., 2018) to evaluate the reconstruction and rendering quality. We additionally report reconstruction metrics such as Accuracy, Completion, Completion Ratio (<7cm) and F1 (<7cm) in the appendix. For GS-based SLAM methods, we also report the number of Gaussians. For semantic segmentation, we report the mean Intersection over Union (mIoU). To evaluate the accuracy of the camera pose, we adopt the average absolute trajectory error (ATE RMSE) (Sturm et al., 2012b).

**Baselines.** We compare our method against several state-of-the-art approaches: NeRF-based SLAM methods, including NICE-SLAM (Zhu et al., 2022a), GO-SLAM (Zhang et al., 2023), and Point-SLAM (Sandström et al., 2023); the semantic NeRF-based SLAM method DNS-SLAM (Li et al., 2023); 3D Gaussian-based SLAM methods such as SplaTAM (Keetha et al., 2024), RTG-SLAM (Peng et al., 2024), and GS-ICP SLAM (Ha et al., 2024); and semantic 3D Gaussian-based SLAM methods, including SGS-SLAM (Li et al., 2024) and OVO-Gaussian-SLAM (Martins et al., 2024). DNS-SLAM, SGS-SLAM and OVO-Gaussian-SLAM are the only semantic SLAM methods available for comparison since the code is not available for other semantic SLAM approaches. Note that SGS-SLAM (Li et al., 2024) and DNS-SLAM (Li et al., 2023) employ test-time optimization using ground truth semantic labels on the test set. SGS-SLAM has been shown to outperform all other existing semantic SLAM methods (Zhu et al., 2024; Li et al., 2023). To ensure a fair comparison and simulating SLAM applications in real-world scenarios where ground truth semantic labels are unavailable, we trained a 2D segmentation model using a Swin backbone (Liu et al., 2021) with Mask2Former (Cheng et al., 2022) on ScanNet, following the same training strategy as our model, and used predicted semantic labels to supervise SGS-SLAM.

## 4.2 RESULTS

**Rendering and Reconstruction Performance.** In Table 1, we evaluate the rendering and reconstruction performance of our method on ScanNet. This is a difficult task compared to the synthetic data where neural RGB-D SLAM methods usually show strong results, because inevitably inaccurate ground truth camera poses and depths make optimization much harder than completely clean synthetic datasets. Compared to existing dense neural RGB-D SLAM methods, our approach achieves state-of-the-art performance on PSNR, SSIM, and Depth L1 metrics. Specifically, our method surpasses the runner-up, DNS-SLAM (Li et al., 2023), by 3.8 dB in PSNR (a 18.6% percent improvement). Furthermore, our approach utilizes approximately **10x** fewer Gaussians than the baselines. This efficiency highlights the effectiveness of our method in achieving high-quality scene representation with reduced computational complexity.

In Table 1, we ran our method with $240 \times 320$ input to compare against GO-SLAM which shares the same tracking method as ours but renders at the same low resolution. GS4 maintains the same PSNR and depth prediction quality as its high resolution version and significantly outperforms GO-SLAM across all metrics.

Fig. 4 shows visual results of RGB and depth rendering. Our method demonstrates superior performance than other GS-SLAM methods. Notably, sometimes the depth maps of our approach even turn out to be better than the noisy ground truth depth inputs. For instance, in the first two columns, our method delivers a more contiguous and complete rendering of the bicycle tires. Similarly, in the middle two columns, we reconstruct the chair's backrest nearly entirely, whereas the GT depth data lacks this detail.

**Semantic Performance.** In Table 2, we present both 2D rendering and 3D mean Intersection over Union (mIoU) scores across the six ScanNet test scenes. For 3D mIoU evaluation, we first align the reconstructed map with the ground-truth mesh and then use 3D neighborhood voting to assign predicted labels. Our method outperforms the previous runner-up, OVO-Gaussian-SLAM, by 19.52% in 3D mIoU. In terms of 2D mIoU, our method surpasses the state-of-the-art semantic Nerf-based SLAM approach, DNS-SLAM, by 17.01%. Qualitative comparisons are included in the appendix.

Table 1: **Rendering Performance on ScanNet.** Values are averaged across the test scenes. Best results are highlighted as first, second. GS Num represents the number of 3D Gaussians included in the scene after mapping is complete.

| Res | Method | PSNR↑ | SSIM↑ | LPIPS↓ | Depth L1↓ | GS Num↓ |
|---|---|---|---|---|---|---|
| 640 × 480 | NICE-SLAM | 17.54 | 0.621 | 0.548 | - | - |
| | Point-SLAM | 19.82 | 0.751 | 0.514 | - | - |
| | DNS SLAM | 20.46 | **0.932** | **0.209** | 6.75 | - |
| | SplaTAM | 18.99 | 0.702 | 0.364 | 7.21 | 2466k |
| | RTG SLAM | 12.75 | 0.372 | 0.761 | 97.56 | 1229k |
| | GS-ICP SLAM | 14.73 | 0.645 | 0.684 | 103.31 | 2565k |
| | SGS SLAM | 15.89 | 0.594 | 0.615 | 11.83 | 2114k |
| | **GS4 (Ours, AFF, 1 iter)** | 22.61 | 0.850 | 0.335 | 6.55 | 356k |
| | **GS4 (Ours, AFF, 5 iters)** | **24.26** | 0.879 | 0.304 | **4.98** | **245k** |
| 320 × 240 | GO-SLAM | 18.21 | 0.657 | 0.553 | 18.14 | - |
| | **GS4 (Ours, AFF, 1 iter)** | 22.50 | 0.885 | 0.238 | 6.04 | 172k |
| | **GS4 (Ours, AFF, 5 iters)** | **24.02** | **0.915** | **0.201** | **5.05** | **130k** |

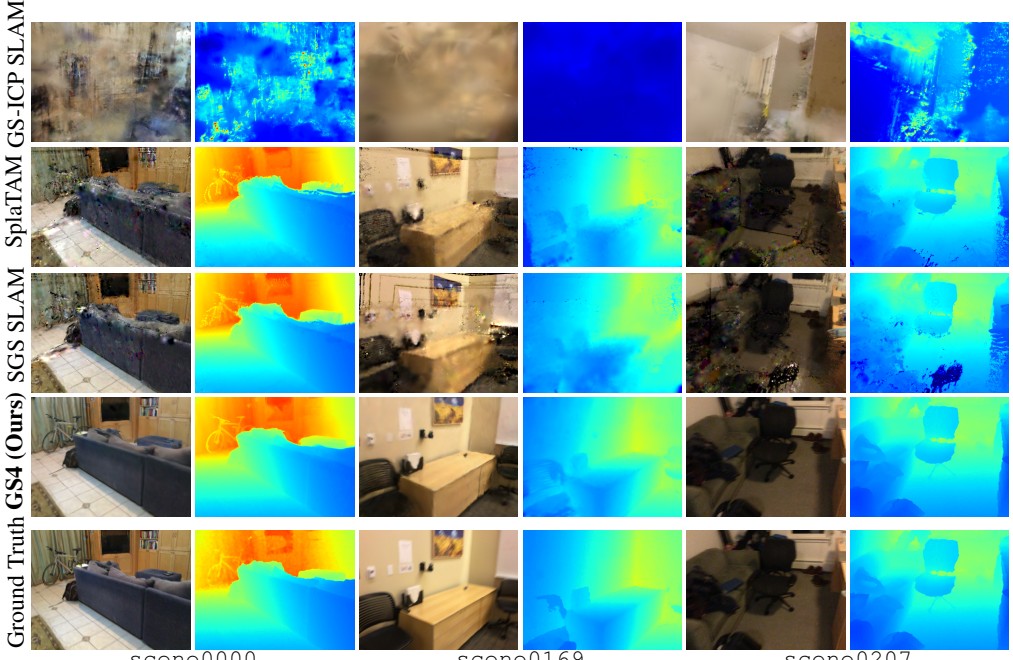

Figure 4: **Renderings on ScanNet**. Our method, **GS4**, renders color & depth for views with fidelity significantly better than all approaches.

**Tracking Performance.** Table 3 shows the tracking results. Our method uses the same tracking algorithm as GO-SLAM, which is significantly better than other GS-based SLAM methods.

**Runtime Comparison.** Table 4 presents a runtime comparison of our method against the baselines at the 640 × 480 resolution, conducted on an Nvidia RTX TITAN. FPS is calculated by dividing the total number of frames by the total time to represent the overall system performance. While GS-ICP SLAM is faster than ours, its rendering and tracking performance is significantly worse (Table 1 and 3). Our approach is **12x** faster than SplaTAM and **17x** faster than SGS-SLAM and DNS-SLAM.

Table 2: **Semantic Performance across ScanNet Test Scenes**

| Methods | DNS SLAM | SGS SLAM | OVO-Gaussian-SLAM | GS4 (Ours, AFF, 1 iter) | GS4 (Ours, AFF, 5 iters) |
|---|---|---|---|---|---|
| mIoU(2D) | 46.03 | 37.20 | ✗ | **63.04** | 62.36 |
| mIoU(3D) | ✗ | 18.87 | 32.58 | **52.10** | 51.89 |

Table 3: **Tracking Performance on ScanNet Test Scenes**. The average values are reported. GS4 uses the same tracking algorithm as GO-SLAM hence the numbers are almost the same.

| Metric | NICE-SLAM | Point-SLAM | DNS-SLAM | SplaTAM | RTG SLAM | GS-ICP SLAM | SGS SLAM | GO-SLAM | **GS4 (Ours)** |
|---|---|---|---|---|---|---|---|---|---|
| ATE RMSE [cm]↓ | 10.70 | 12.19 | 48.07 | 11.88 | 144.52 | NaN | 40.97 | **7.00** | **6.98** |

Table 4: **Average Runtime on ScanNet Test Scenes**

| Methods | Point-SLAM | DNS-SLAM | SplaTAM | RTG-SLAM | GS-ICP SLAM | SGS-SLAM | GS4 (ours, AFF, 1 iter) | GS4 (ours, AFF, 5 iters) |
|---|---|---|---|---|---|---|---|---|
| FPS ↑ | 0.05 | 0.18 | 0.23 | 1.01 | **3.62** | 0.17 | 2.87 | 1.92 |

**Zero-shot Experiments.** In Table 5, we report quantitative zero-shot results. For NYUv2, the numbers are averaged over three scenes, and for TUM-RGBD, they are also averaged over three scenes. Per-scene results are provided in the appendix. On NYUv2, our method outperforms all other GS-based SLAM approaches across all rendering metrics while using significantly fewer Gaussians. Qualitative comparisons are also provided in the appendix. On TUM-RGBD, our method outperforms the baselines in terms of SSIM and the number of Gaussians, and closely matches the best performance in other metrics, despite relying primarily on a feed-forward model trained on ScanNet.

Table 5: **Zero-shot Rendering Performance on NYUv2 and TUM-RGBD.** Values are averaged across the test scenes. GS Num represents the number of 3D Gaussians in the scene after mapping.

| Dataset | Res | Method | PSNR↑ | SSIM↑ | LPIPS↓ | GS Num↓ |
|---|---|---|---|---|---|---|
| NYUv2 | $640 \times 480$ | SplaTAM | 18.86 | 0.692 | 0.372 | 1236k |
| | | RTG-SLAM | 11.84 | 0.221 | 0.703 | 807k |
| | | SGS-SLAM | 19.32 | 0.708 | 0.357 | 1108k |
| | | **GS4 (Ours, AFF, 5 iters)** | **22.09** | **0.853** | **0.268** | **278k** |
| TUM RGBD | $640 \times 480$ | SplaTAM | **22.76** | 0.891 | **0.182** | 803k |
| | | RTG-SLAM | 19.75 | 0.769 | 0.395 | 198k |
| | | SGS-SLAM | 22.44 | 0.876 | 0.184 | 735k |
| | | **GS4 (Ours, AFF, 5 iters)** | 22.70 | **0.903** | 0.191 | **166k** |

**Ablation Study.** We conduct an ablation study using all ScanNet test scenes, as shown in Table 6. The results demonstrate that both the Gaussian Refinement Network and the Few-Iteration Gaussian Optimization are critical to the performance of GS4. Additionally, Gaussian pruning significantly reduces the number of Gaussians without sacrificing accuracy. More results on backbone comparisons, reconstruction accuracy and qualitative results are shown in the appendix.

Table 6: **Ablation on ScanNet (averaged over test scenes)**

| Design Choice | PSNR [dB]↑ | SSIM↑ | LPIPS↓ | Depth L1↓ | mIoU↑ | Gs Num↓ |
|---|---|---|---|---|---|---|
| GS Prediction | 14.91 | 0.460 | 0.663 | 33.14 | 41.0 | 133k |
| + GS Refinement | 16.15 | 0.556 | 0.584 | 29.56 | 44.68 | 576k |
| + Few-Iter. Optimization (1) | 22.66 | 0.852 | 0.335 | 6.41 | 63.7 | 680k |
| + GS Pruning (Full SLAM) | 22.61 | 0.851 | 0.335 | 6.56 | 63.1 | 355k |

## 5 CONCLUSION

We present GS4, a novel SLAM system that incrementally constructs and updates a 3D semantic scene representation from a monocular RGB-D video with a learned generalizable network. Our novel Gaussian refinement network and few-iteration Gaussian optimization significantly improve the performance of our approach. Our experiments demonstrate state-of-the-art semantic SLAM performance on the ScanNet benchmark while running 10x faster and using 10x less Gaussians than baselines. The model also showed strong generalization capabilities through zero-shot transfer to the NYUv2 and TUM RGB-D datasets. In future work, we will further improve the computational speed of GS4 and explore options for a pure RGB-based SLAM approach.

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

# Appendix

## A MORE EXPERIMENTAL SETUP

For ScanNet, we evaluate on six scenes (scene0000, scene0059, scene0106, scene0169, scene0181, and scene0207), which are commonly used by other SLAM methods.

## B ALTERNATIVE BACKBONE

In Table 7, we present the average rendering metrics across all ScanNet test scenes using different backbones for the prediction model. It can be seen that our method works effectively with different backbones.

Table 7: Comparison between Swin and AFF backbones on different metrics.

| Backbone | PSNR | SSIM | LPIPS | Depth L1 | mIoU | GS Num |
|---|---|---|---|---|---|---|
| GS4 (Swin, 1 iter) | 22.50 | 0.846 | 0.356 | 6.338 | 64.4 | 361k |
| GS4 (AFF, 1 iter) | 22.61 | 0.851 | 0.335 | 6.558 | 63.0 | 355k |

## C RECONSTRUCTION RESULTS ON SCANNET

In Table 8, we present the reconstruction metrics on the ScanNet dataset. GS4 outperforms all baselines in terms of completion and F-score.

| **Methods** | Acc. ↓ | Comp. ↓ | Comp. Ratio (<7cm) ↑ | F-Score (<7cm) ↑ | GS Num↓ |
|---|---|---|---|---|---|
| SplaTAM | **8.10** | 5.58 | 76.34 | 75.95 | 2466k |
| RTG-SLAM | 99.80 | 47.44 | 24.61 | 16.69 | 1229k |
| SGS-SLAM | 17.11 | 13.75 | 55.01 | 55.26 | 2114k |
| GS4 | 8.56 | **3.88** | **87.34** | **79.90** | **295k** |

Table 8: Reconstruction metrics on ScanNet

## D QUALITATIVE COMPARISON FOR SEMANTIC SEGMENTATION.

As illustrated in Fig. 5, our approach achieves superior semantic segmentation accuracy compared to the SGS-SLAM baseline. For example, in the first column of Fig.5, our semantic rendering provides a more accurate representation of the desks, chairs, and night tables than SGS-SLAM.

## E ZERO-SHOT RESULTS ON NYUV2

Fig. 6 illustrates our zero-shot visualization results on the NYUv2 dataset. Despite our models being exclusively trained on the ScanNet dataset, our method demonstrates superior performance on the NYUv2 dataset compared to other GS-based SLAM approaches. In Table 9, we present the quantitative zero-shot results across three scenes from the NYUv2 dataset. Our method outperforms all other GS-based SLAM approaches on all rendering metrics, while using significantly fewer Gaussians.

## F ZERO-SHOT RESULTS ON TUM RGB-D

In Table 10, we present the quantitative zero-shot results across three scenes from the TUM RGB-D dataset. TUM RGB-D provides ground truth camera trajectories, so we also report tracking per-

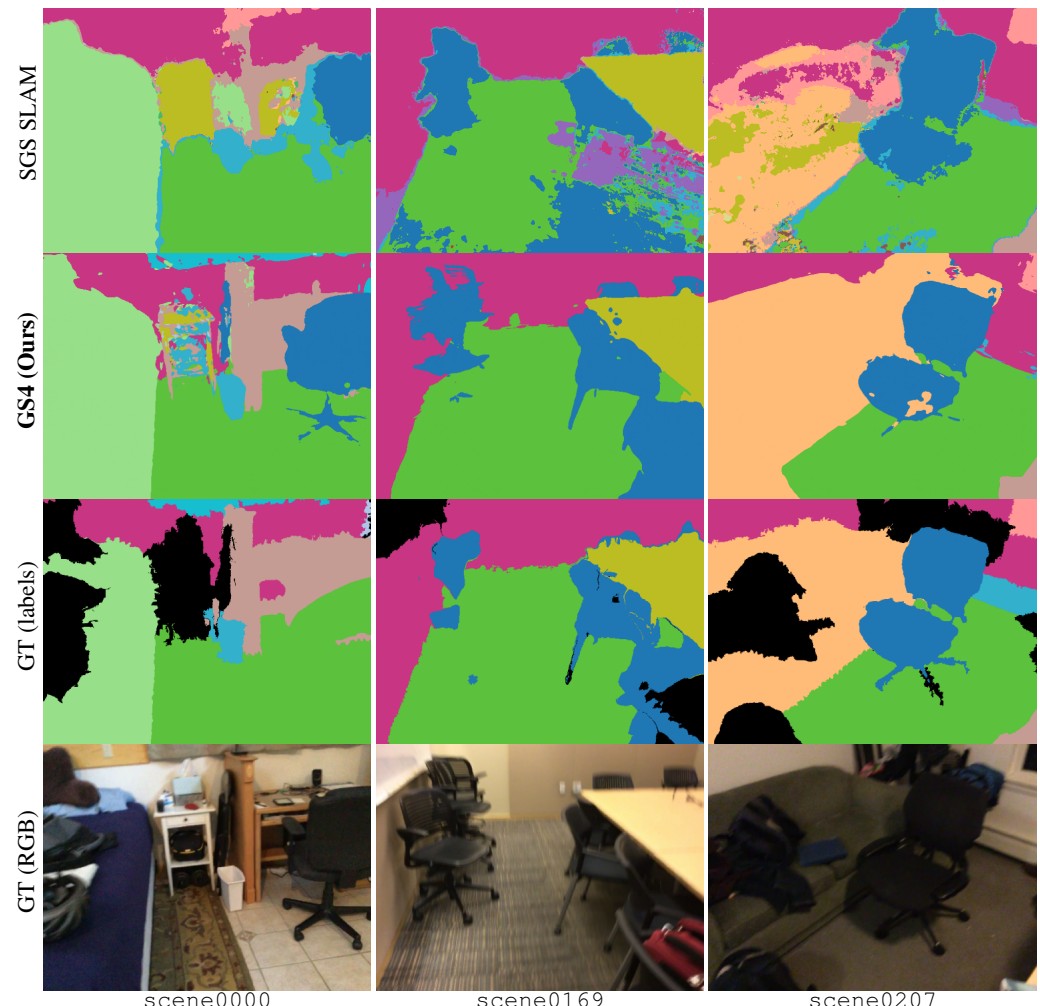

Figure 5: **Semantic Renderings on ScanNet**. Qualitative comparison on semantic synthesis of our method and baseline semantic SLAM method SGS-SLAM. Black areas in GT labels denote regions that are unannotated.

formance. Our method achieves rendering performance comparable to that of all other GS-based SLAM approaches, while using significantly fewer Gaussians.

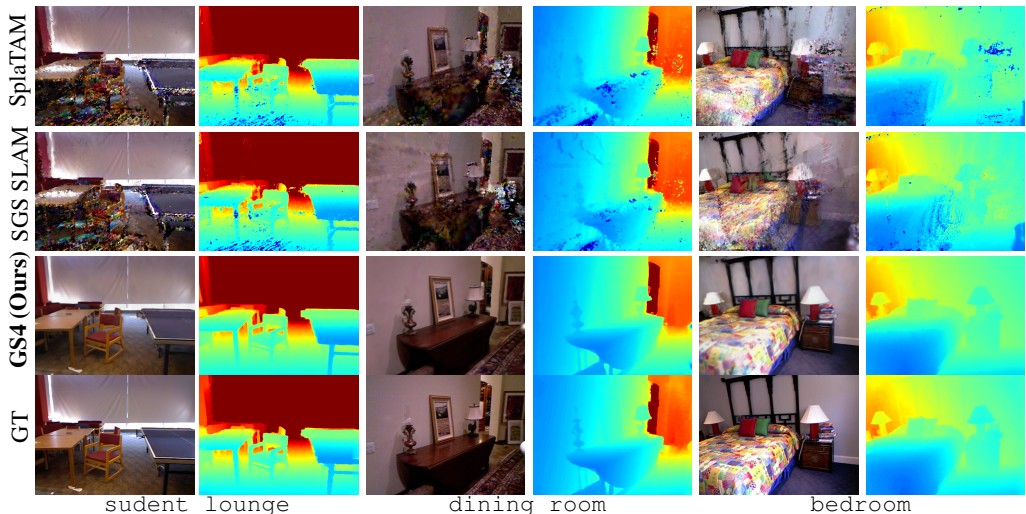

Figure 6: **Zero-shot Visualization on NYUv2**. Qualitative comparison of our method and other GS-based SLAM methods.

Table 9: Rendering and Runtime performance on NYUv2 test scenes with $640 \times 480$ input. GS Num represents the number of 3D Gaussians included in the scene after mapping is complete. FPS is conducted on an Nvidia RTX TITAN.

| Methods | Metrics | bedroom | student lounge | dining room | Avg |
|---|---|---|---|---|---|
| SplaTAM | PSNR↑ | 17.99 | 20.77 | 17.82 | 18.86 |
| | SSIM↑ | 0.692 | 0.795 | 0.589 | 0.692 |
| | LPIPS↓ | 0.343 | 0.309 | 0.465 | 0.372 |
| | GS Num↓ | 1529k | 1116k | 1063k | 1236k |
| RTG-SLAM | PSNR↑ | 10.81 | 12.94 | 11.76 | 11.84 |
| | SSIM↑ | 0.146 | 0.299 | 0.217 | 0.221 |
| | LPIPS↓ | 0.738 | 0.662 | 0.709 | 0.703 |
| | GS Num↓ | 906k | 591k | 925k | 807k |
| SGS-SLAM | PSNR↑ | 19.66 | 20.41 | 17.90 | 19.32 |
| | SSIM↑ | 0.754 | 0.780 | 0.590 | 0.708 |
| | LPIPS↓ | 0.289 | 0.318 | 0.463 | 0.357 |
| | GS Num↓ | 1201k | 1074k | 1049k | 1108k |
| GS4 (Ours) | PSNR↑ | **20.86** | **20.59** | **22.82** | **22.09** |
| | SSIM↑ | **0.870** | **0.862** | **0.829** | **0.854** |
| | LPIPS↓ | **0.230** | **0.251** | **0.322** | **0.268** |
| | GS Num↓ | **257k** | **206k** | **371k** | **278k** |

Table 10: Rendering, Tracking, and Runtime performance on TUM RGB-D test scenes with $640 \times 480$ input. GS Num represents the number of 3D Gaussians included in the scene after mapping is complete. FPS is conducted on an Nvidia RTX TITAN.

| Methods | Metrics | fr1_desk | fr2_xyz | fr3_office | Avg |
|---|---|---|---|---|---|
| SplaTAM | PSNR↑ | 22.07 | 24.66 | 21.54 | **22.76** |
| | SSIM↑ | 0.857 | **0.947** | 0.870 | 0.891 |
| | LPIPS↓ | 0.238 | 0.099 | **0.210** | **0.182** |
| | ATE RMSE↓ | 3.33 | 1.55 | 5.28 | 3.39 |
| | GS Num↓ | 969k | 635k | 806k | 803k |
| RTG-SLAM | PSNR↑ | 18.49 | 20.18 | 20.59 | 19.75 |
| | SSIM↑ | 0.715 | 0.795 | 0.797 | 0.769 |
| | LPIPS↓ | 0.438 | 0.353 | 0.394 | 0.395 |
| | ATE RMSE↓ | **1.66** | **0.38** | **1.13** | **1.06** |
| | GS Num↓ | 236k | **84k** | 273k | 198k |
| SGS-SLAM | PSNR↑ | **22.10** | **25.61** | 19.62 | 22.44 |
| | SSIM↑ | **0.886** | 0.946 | 0.796 | 0.876 |
| | LPIPS↓ | **0.176** | **0.097** | 0.280 | 0.184 |
| | ATE RMSE↓ | 3.57 | 1.29 | 9.08 | 4.65 |
| | GS Num↓ | 808k | 695k | 701k | 735k |
| GS4 (Ours) | PSNR↑ | 21.71 | 23.86 | **22.54** | 22.70 |
| | SSIM↑ | 0.877 | 0.904 | **0.890** | **0.903** |
| | LPIPS↓ | 0.242 | 0.154 | 0.226 | 0.191 |
| | ATE RMSE↓ | 1.86 | 0.63 | 1.95 | 1.48 |
| | GS Num↓ | **175k** | 87k | **190k** | **166k** |

