# OpenReview forum: "GS4: Generalizable Sparse Splatting Semantic SLAM"
_ICLR.cc/2026/Conference — ICLR 2026 Conference Withdrawn Submission_

### Official Review · Reviewer_oZoD · 2025-10-18

**Soundness:** 3
**Presentation:** 3
**Contribution:** 3
**Rating:** 6
**Confidence:** 5

**Summary:**

GS4 proposes a generalizable, feed‑forward semantic SLAM system built on 3D Gaussian Splatting for RGB‑D video. Instead of per‑scene optimization, it directly predicts 3D Gaussians (position, scale, rotation, color, opacity) and their semantic labels from each frame using a transformer-based backbone, then incrementally fuses them into a global map. A learned Gaussian refinement network replaces heuristic densification/pruning by merging overlapping Gaussians across frames and pruning low‑utility ones, yielding a compact representation. To handle pose corrections from loop closure and bundle adjustment, GS4 performs a few iterations of rendering‑based optimization to fix drift and suppress floaters, keeping computation low while improving map fidelity.

Empirically, GS4 achieves state‑of‑the‑art rendering, depth, and semantic performance on ScanNet while using roughly an order of magnitude fewer Gaussians than prior GS‑SLAM methods, with competitive tracking accuracy and substantially higher throughput. It further shows zero‑shot generalization to NYUv2 and TUM RGB‑D.

Its main contributions are a unified, generalizable Gaussian prediction framework that jointly handles mapping and semantics; a learned Gaussian refinement/merging module that produces sparse, high-quality maps without per-scene optimization; and a lightweight, few-iteration post-BA optimization that effectively corrects misalignments with minimal cost, validated through strong results and ablations.

**Strengths:**

While prior Gaussian Splatting (GS) SLAM systems (e.g., SplaTAM, SGS-SLAM) rely on per-scene optimization, GS4 replaces this with a feed-forward, generalizable network that predicts semantic Gaussians directly from RGB-D inputs. This shifts the paradigm from scene-specific reconstruction to learned, zero-shot mapping—a significant conceptual leap.

Unlike prior semantic SLAM methods that pipeline segmentation models (e.g., Mask2Former) separately from geometry estimation, GS4 unifies color, depth, and semantic prediction within a single transformer-based architecture. This avoids error propagation and enables end-to-end training.

 Instead of heuristic densification/pruning (as in 3DGS or SplaTAM), GS4 introduces a Gaussian Refinement Network that learns to merge Gaussians across frames in a differentiable, data-driven way. This is a novel application of local-attention transformers to 3D primitive consolidation.

**Weaknesses:**

While the paper claims zero-shot semantic generalization to NYUv2 and TUM RGB-D, these datasets share the same indoor domain and similar class taxonomy (e.g., chairs, desks, walls) as ScanNet. There is no evaluation on domain-shifted semantics, such as outdoor scenes (e.g., KITTI with vehicle/pedestrian classes) or open-vocabulary settings (e.g., using CLIP embeddings as in OVO-SLAM).

While GS4 is “10x faster” than baselines, FPS numbers (Table 4) hide critical details: Is speedup due to fewer Gaussians, no per-scene optimization, or both? What is the breakdown of runtime (prediction vs. refinement vs. optimization)?

Experiments are limited to short ScanNet sequences (~2,680 frames avg). Real-world SLAM must handle hours-long trajectories with revisits, dynamic objects, and lighting changes.

**Questions:**

See the Weaknesses.

---

### Official Review · Reviewer_DAQQ · 2025-10-28

**Soundness:** 3
**Presentation:** 3
**Contribution:** 2
**Rating:** 4
**Confidence:** 4

**Summary:**

This paper introduces a generalizable Gaussian splatting module that integrates semantic information into a SLAM system. The core idea is to employ a generalizable Gaussian model to predict Gaussians enriched with semantic information directly from the input RGB image, depth map, and estimated camera pose—rather than optimizing the Gaussians during tracking, as done in recent 3D Gaussian-based SLAM methods. Experimental results on zero-shot datasets demonstrate the effectiveness and generalization capability of the proposed SLAM system.

**Strengths:**

1. The paper is well-organized and easy to follow.
2. Introducing a generalizable 3D Gaussian module into SLAM is a promising idea, as it can reduce mapping time compared to other 3DGS-based SLAM systems.
3. The proposed method effectively reduces the total number of 3D Gaussians required to represent the entire scene while maintaining high rendering performance.

**Weaknesses:**

1. This SLAM module relies on the state-of-the-art tracking component from GO-SLAM, which may lead to an unfair comparison with other 3DGS-based SLAM methods. For instance, SplaTAM performs both tracking and mapping using its learned 3D Gaussian map. Therefore, it would be important to clarify whether other 3DGS-based methods could also achieve improved rendering performance if they adopted the same tracking module from GO-SLAM.
2. The proposed SLAM framework primarily focuses on enhancing mapping performance. However, the improved mapping does not appear to benefit the tracking thread, which limits the overall contribution to the SLAM domain. Consequently, this module might be more appropriately evaluated within the context of 3D reconstruction rather than as a complete SLAM system.
3. Regarding semantic information, it would be beneficial to demonstrate the effectiveness of incorporating semantic cues into the generalizable model to highlight their contribution to performance improvement.

**Questions:**

Please refer to the weakness part.

---

### Official Review · Reviewer_i2AF · 2025-10-28

**Soundness:** 2
**Presentation:** 2
**Contribution:** 3
**Rating:** 4
**Confidence:** 5

**Summary:**

This paper proposes GS4 (Generalizable Sparse Splatting Semantic SLAM), a novel SLAM system that integrates 3D Gaussian Splatting (3DGS) into semantic visual SLAM while achieving both generalization and efficiency. Unlike previous Gaussian-based SLAM methods that require per-scene gradient optimization, GS4 introduces a feed-forward architecture capable of directly predicting a sparse set of 3D semantic Gaussians from RGB-D input frames.

**Strengths:**

1. The paper presents strong innovation in reconstruction. It provides a new perspective on combining feed-forward models and SLAM, which is conceptually inspiring and technically interesting.
2. The overall structure is clear and easy to follow. The paper is well-organized and logically written, making the technical content accessible even to readers not deeply familiar with Gaussian Splatting.
3. The writing quality is fluent and professional, with clear English exposition throughout.

**Weaknesses:**

1. Experimental evaluation is relatively weak.
The main experiments are conducted on only six test scenes from the ScanNet dataset. This makes it hard to convincingly demonstrate the model’s effectiveness and generalization. It would be more appropriate to adopt standard benchmarks commonly used for 3DGS-based SLAM systems.
In addition, the experiments on NYUv2 and TUM RGB-D are labeled as zero-shot, which may not be the most suitable term for a SLAM system. The use of this term feels inconsistent with standard SLAM terminology.
For pose estimation, the ScanNet ATE improvement over GO-SLAM is only 0.02 cm, which is negligible. If camera tracking is intended as part of the contribution, results on standard 3DGS-SLAM localization benchmarks would be necessary.

2. Experimental setup lacks detailed description.
It is unclear whether the reconstruction metrics are computed on train views or test views, and how these views are defined (e.g., whether keyframes are considered train views). More explicit clarification of evaluation protocols is needed.
3. On the number of Gaussians.
The paper emphasizes that GS4 uses 10× fewer Gaussians. However, fewer Gaussians are not necessarily better — in practice, a more compact representation may lead to higher risk of reconstruction artifacts or missing structures in unseen views. The evaluation metrics used here may not fully capture such trade-offs.
4. Semantic evaluation is insufficient.
While the paper claims contributions in semantic Gaussian SLAM, the experiments and discussion on the semantic component are relatively limited. The semantic performance section could be expanded with more analysis, visualizations, and ablations.

**Questions:**

Please refer to the issues mentioned above — especially regarding the evaluation settings, benchmark choice, and interpretation of Gaussian sparsity.

---

### Official Review · Reviewer_yZzb · 2025-10-28

**Soundness:** 2
**Presentation:** 3
**Contribution:** 2
**Rating:** 4
**Confidence:** 4

**Summary:**

The paper proposes GS4, a generalizable 3DGS–based semantic SLAM framework. GS4 introduces a feed-forward Gaussian Prediction Model and a Gaussian Refinement Network that jointly infer and merge 3D semantic Gaussians from RGB-D frames, elminating the need for per-scene optimization. A lightweight few-iteration optimization step further refines the map to correct drift and artifacts. Experiments on the ScanNet benchmark show competitive performance while using ten times fewer Gaussians and running faster than baseline methods, with zero-shot generalization capability to NYUv2 and TUM RGB-D datasets.

**Strengths:**

- The paper is well written and structured. The ablation study clearly shows the contribution of each module

- GS4 removes the need for per-scene optimization by introducing a feed-forward Gaussian prediction and refinement pipeline, which simplifies deployment and enables real-time performance on standard RGB-D inputs.

- The proposed refinement and pruning strategy effectively maintains scene quality while reducing the number of Gaussians by roughly one order of magnitude compared to existing GS-SLAM systems, showing practical efficiency gains.

**Weaknesses:**

- The terminology “monocular RGB-D” is confusing. The paper repeatedly refers to “monocular RGB-D sequences,” which is inconsistent with the common definition of a monocular system that does not include depth sensing. In fact, datasets such as TUM RGB-D use the Microsoft Kinect v1 sensor, where depth is computed via a stereo setup. For clarity and technical correctness, it is recommended to simply use the term “RGB-D system” throughout the paper.

- Tracking evaluation does not reflect GS4’s contribution. As stated under Table 3, GS4 adopts the same tracking module as GO-SLAM, resulting in nearly identical tracking results. This makes the novelty of the tracking part limited.

- Incomplete zero-shot evaluation baselines. Although OVO-SLAM is listed as one of the semantic baselines, the zero-shot experiments on NYUv2 and TUM RGB-D (Table 5 and Appendix) only compare against SplaTAM, RTG-SLAM, and SGS-SLAM, not including OVO-SLAM. As OVO-SLAM is a recent open-vocabulary semantic SLAM method, it should be included in comparisons.

- Insufficient details in 3D semantic evaluation protocol. The paper reports 3D mIoU by aligning reconstructed maps with ground-truth meshes and applying 3D neighborhood voting. However, details are missing, such as the class set used, label mapping strategy, and treatment of unlabeled (“black”) regions. In particular, when comparing closed-set and open-vocabulary methods (e.g., OVO-SLAM), the label-space harmonization and handling of unannotated areas should be clearly explained.

- Missing recent baselines in quantitative results. The related work section discusses MonoGS and other strong GS-based reconstruction systems, but these methods are absent from the main quantitative tables (Table 1, 5, 8). Even if they are not semantic, including such recent GS-SLAM variants, including MonoGS, SplaSLAM.

- Need some discussion of recent feedforward online reconstruction methods, such as Mast3RSLAM, SLAM3R, and the streaming reconstruction methods such as CAT3R, Streaming VGGT etc.

**Questions:**

The questions/concerns are provided in weakness.

---

### Note · Authors · 2025-11-13

I have read and agree with the venue's withdrawal policy on behalf of myself and my co-authors.